# Quality Evaluation of Atractylodis Macrocephalae Rhizoma Based on Combinative Method of HPLC Fingerprint, Quantitative Analysis of Multi-Components and Chemical Pattern Recognition Analysis

**DOI:** 10.3390/molecules26237124

**Published:** 2021-11-25

**Authors:** Cheng Zheng, Wenting Li, Yao Yao, Ying Zhou

**Affiliations:** NMPA Key Laboratory of Quality Evaluation of Traditional Chinese Medicine (Traditional Chinese Patent Medicine), Zhejiang Institute for Food and Drug Control, Hangzhou 310052, China; zhengcheng@zjyj.org.cn (C.Z.); liwenting@zjyj.org.cn (W.L.); yaoyao@zjyj.org.cn (Y.Y.)

**Keywords:** Atractylodis Macrocephalae Rhizoma, fingerprint, content determination, chemical pattern recognition analysis, quality evaluation

## Abstract

A method for the quality evaluation of Atractylodis Macrocephalae Rhizoma (AMR) based on high-performance liquid chromatography (HPLC) fingerprint, HPLC quantification, and chemical pattern recognition analysis was developed and validated. The fingerprint similarity of the 27 batches of AMR samples was 0.887–0.999, which indicates there was very limited variance between the batches. The 27 batches of samples were divided into two categories according to cluster analysis (CA) and principal component analysis (PCA). A total of six differential components of AMR were identified in the partial least-squares discriminant analysis (PLS-DA), among which atractylenolide I, II, III, and atractylone counted 0.003–0.045%, 0.006–0.023%, 0.001–0.058%, and 0.307–1.175%, respectively. The results indicate that the quality evaluation method could be used for quality control and authentication of AMR.

## 1. Introduction

Atractylodis Macrocephalae Rhizoma (AMR), named “Baizhu” or “Zhu”, is a member of Atractylodes and one of the “Eight Famous Herbs from Zhejiang Province”. AMR was first documented in “Shennong Ben Cao Jing” (also known as “The Classic of Herbal Medicine”), where it is classified as “noble” or “upper herbs”. AMR is the dried rhizome of Atractylodis Macrocephalae, with a fresh fragrance and a sweet, slightly pungent taste. Its functions include invigorating the spleen and replenishing qi, drying dampness and diuresis, antiperspirant, and anti-fetus. It is used for treating spleen deficiency, no appetite, abdominal distension, diarrhea, phlegm, palpitations, edema, spontaneous perspiration, and fetal irritability [1]. Several previous studies have reported that AMR has various pharmacological activities such as immune enhancement [2], anti-tumor [3,4], neuroprotection [5], anti-oxidative [6,7], anti-inflammatory, etc. [8,9,10]. Most of the researchers have previously shown that chemical components of AMR are focused on sesquiterpenes, particularly atractylenolide I, II, III, and atractylone [11,12,13,14,15,16].

High-performance liquid chromatography (HPLC) fingerprinting has become the most widely used method because of its efficiency and convenience. Some scholars have also conducted exploratory work on the fingerprint of AMR of different origins [17,18]. Studies on the quantitative analysis of two or more ingredients of AMR including atractylenolide I, II, III, and atractylone, partial combined principal component analysis (PCA) have been reported frequently [19,20,21]. However, little research has been conducted on the fingerprint profiling of “Raw AMR” and “Stir-Baked AMR with Bran” coupled with quantification of four ingredients and characterization by chemometric analysis including cluster analysis (CA), principal component analysis (PCA), and partial least-squares discriminant analysis (PLS-DA). Meanwhile, the corresponding limits have not been established so far. Therefore, the present study aimed to investigate the HPLC fingerprint of AMR and quantitative analysis of atractylenolide I, II, III, and atractylone on the basis of chemometric analysis. Additionally, in the present investigation, we established reasonable limits that have certain emergent novelty and very high practical value.

## 2. Results and Discussion

### 2.1. Selection of Detection Wavelength

The full wavelength of AMR solution was scanned with an ultraviolet detector. In order to make the fingerprints informative and ensure that the multiple components do not interfere with each other at the maximum absorption wavelength, 235 nm was chosen as the detection wavelength in the final fingerprint. The detection wavelength of atractylenolide I was 280 nm, and the detection wavelength of atractylenolide II, atractylenolide III, and atractylone was 220 nm for multi-indicator content determination (Appendix A).

### 2.2. Optimization of Chromatographic Conditions

The separation effects of different columns (Eclipse plus C18 and Kromasil 100-5C18), as well as different mobile phases (acetonitrile—0.1% formic acid and acetonitrile-water), different flow rates (1.0 mL·min^−1^ and 0.9 mL·min^−1^), different sampling volumes (5 µL, 10 µL, and 20 µL) were compared to evaluate the fitness of chromatographic conditions. It was found that the separation resolution of AMR was best under Kromasil 100-5C18 column with acetonitrile-water with gradient elution as the mobile phase, a flow rate of 1.0 mL·min^−1^, and sample volume of 20 µL, so these were set as the optimum chromatographic conditions for further use.

### 2.3. Identification of the Common Peaks

The HPLC fingerprints generated by the 27 batches of AMR were analyzed and 4 common peaks were found (Figure 1). Among them, four common peaks were identified using the developed HPLC method based on the comparison of their retention time with the reference substances. Peaks 2, 6, 18, and 14 were identified as atractylenolide III, atractylenolide II, atractylenolide I, and atractylone, respectively. Some batches of AMR with low content of atractylenolide I (peak 18) were not scanned at 235 nm, therefore not in the scope of the HPLC fingerprint research.

Sesquiterpenes play an extremely important role in biological activity among the main chemical components of AMR [8,16]. In view of the results of CA, PCA, and PLS-DA, the main marker components between raw AMR and AMR stir-fried with bran were basically consistent with the above common peaks.

### 2.4. Validation of the HPLC Fingerprint Method

For the precision study, the retention time and the peak area of peak 14 (atractylone) were chosen as the reference, and the relative retention time (RRT) and the relative peak area (RPA) of the 17 common peaks of all the samples were calculated. The relative standard deviations (RSDs) of the RPA of each common peak were found to be less than 2.0%, and RRT of each common peak was found to be less than 1.0% (Table 1), which showed that the precision of the HPLC fingerprint method was good.

The RPA and RRT of the 17 common peaks were calculated in the repeatability test. The RSDs of the RRT for each peak were less than 0.9%. The RSDs of the RPA were found to be less than 2.2%. The two RSDs indicated that the repeatability of the HPLC method was satisfactory.

For the stability test, the sample solution was measured at 0, 2, 4, 8, 10, 16, and 24 h after preparation, and then the RRT and the RPA were calculated. The RSD of the RRT was found to be less than 0.7%. The RSD of the RPA was found to be less than 2.5%. The results showed that the AMR sample solution was stable within 24 h.

The chromatographic profiles of 16 batches of raw AMR (S12–S27) samples were processed and a simulative mean chromatogram was generated. The similarities of different chromatographic fingerprints were compared with the simulative mean chromatogram. The similarity of 27 batch samples ranged from 0.887 to 0.999 (Table 2, Figure 2), showing that the chemical composition consistency of different processing methods in different batches of AMR was not very good.

### 2.5. Chemical Pattern Recognition Analysis

#### 2.5.1. Systematic Cluster Analysis

The relative peak areas of 17 common peaks in the HPLC chromatogram fingerprint of 27 batches of AMR were used as variables, and systematic cluster analysis was performed with SIMCA-P; the results showed that the 27 batches of AMR could be divided into two categories, in which S1–11 were clustered into one category and S12–27 were clustered into the other category (Figure 3), indicating that the quality of AMR and AMR stir-fried with bran was more uniform and stable, respectively.

#### 2.5.2. Principal Component Analysis (PCA)

The peak areas of the common peaks of the 27 batches of AMR were standardized by SIMCA-P software, and the eigenvalues of the correlation matrix were calculated as the variance contribution rate. Taking the feature value greater than 1 as the extraction standard, the cumulative variance contribution of the four principal components was 76.44%, which could basically represent most of the information on the HPLC fingerprint. As indicated by the results, the information on the first principal component was mainly derived from chromatographic peaks 2~5; the information on the second principal component was mainly derived from peaks 1, 2, 4, 6, and 11; the information on the third principal component was mainly derived from peak 8, and the information on the fourth principal component was mainly derived from peaks 9 and 10. The peak areas in 27 batches of AMR and AMR stir-fried with bran were carried on PCA analysis. As seen in Figure 4, AMR and AMR stir-fried with bran could be clearly classified into 2 categories, which was consistent with the results of CA.

#### 2.5.3. Partial Least-Squares Discriminant Analysis (PLS-DA)

Supervised PLS-DA was performed on 27 batches of AMR fingerprint data to establish a PLS-DA model that had good explanatory and predictive power (R2Y = 90.8, Q2 = 74.9). The PLS-DA score plot revealed (Figure 5) that AMR and AMR stir-fried with bran were clearly distributed in two quadrants, indicating significant differences in their chemical composition. Further, variable importance for the projection (VIP) >1 was used as a criterion for screening out the main difference peaks between raw AMR and AMR stir-fried with bran, as shown in Figure 6. Peak 16 (P16), peak 14 (P14), peak 15 (P15), peak 13 (P13), peak 17 (P17), and peak 6 (P6) were the index components of the main differences, which was consistent with the results of the S-plot (Figure 7). Meanwhile, the peak areas of peaks16 and 14 were generally higher in AMR than AMR stir-fried with bran.

### 2.6. Validation of the HPLC Quantification Method

The range of linearity was established by injecting six different concentrations obtained by the dilution of a standard of atractylenolide III, atractylenolide II, atractylenolide I and atractylone. Analytical curves for each compound were obtained considering the correlation between the peak area and the respective concentration of the standard. The linearity data including slope, intercept, and correlation coefficient (R^2^) were calculated, and they are presented in Table 3. As can be seen from the table, the linearity was satisfactory in all cases with correlation coefficients (R^2^ > 0.999). R values for the calibration curves higher than 0.99 verified that the linearity was adequate for the intended purpose.

Good precision, as revealed in the relative standard deviations (RSDs) for peak area of atractylenolide III, atractylenolide II, atractylenolide I, and atractylone, was 0.19%, 0.25%, 0.30%, and 0.13%, respectively.

The repeatability of the method was tested by the determination of a sample of S9. The RSDs for the contents of atractylenolide III, atractylenolide II, atractylenolide I, and atractylone were 2.66%, 2.76%, 2.43%, and 2.41%, respectively.

For the stability test, the RSDs of peak area of atractylenolide III, atractylenolide II, atractylenolide I, and atractylone were 0.34%, 0.45%, 0.33% and 0.11%, respectively, indicating the standard solution was stable for 24 h at ambient temperature.

The recovery test was conducted to evaluate the accuracy of this method. The recovery test of solution was obtained by adding a known amount of atractylenolide III, atractylenolide II, atractylenolide I, and atractylone standard solution, respectively, to the six AMR solutions. As shown in Table 4, the recovery rates for atractylenolide III, atractylenolide II, atractylenolide I, and atractylone were within the range of 87.50% and 106.26%. The RSDs for the recovery rate were within the range of 1.13% and 2.00%.

### 2.7. Quantification of the Samples of AMR

The proposed HPLC method was successfully applied to the quantification of atractylenolide III, atractylenolide II, atractylenolide I, and atractylone in AMR. As shown in Table 5, the contents of atractylenolide II in AMR were within the range of 0.006–0.023%; the contents of atractylenolide I in AMR were within the range of 0.003–0.039%; and the contents of atractylone in AMR were within the range of 0.331–1.175% with a great difference.

### 2.8. Limit Setting

The content of atractylone in AMR stir-fried with bran was generally lower than that in raw AMR, while the contents of atractylenolide I~III showed the opposite results. Therefore, calculated as dry product, the total amount of Atractylodes I~III should not be less than 0.025%, the sum of the four should not be less than 0.6%, the total amount of Atractylodes I~III in AMR stir-fried with bran should not be less than 0.045%, and the sum of the four should not be less than 0.4%. The limit can well distinguish AMR from AMR stir-fried with bran.

## 3. Materials and Methods

### 3.1. Chemicals and Reagents

Atractylenolide I (batch number (BN):111975-201501, mass fraction (MF): 99.9%), Atractylenolide II (BN:111976-201501, MF: 99.9%) and Atractylenolide III (BN: 111978-201501, MF: 99.9%) reference substances were all purchased from China Institute for Food and Drug Control; atractylone (BN: PS011118, MF: 98%, Chengdu Pusi Biotechnology Co., Ltd., Chengdu, China); Acetonitrile, HPLC grade (Merck & Co., Kenilworth, NJ, USA); formic acid, HPLC grade (Sigma Corporation, Cream Ridge, NJ, USA); methanol of analytical purity; ultrapure water, prepared by Milli-Q system.

Atractylodis Macrocephalae Rhizoma (numbered S1 to S27) were purchased from pharmaceutical factories and Chinese medicine hospitals. The samples were collected from Zhejiang and Anhui of China for analysis, and the source information is listed in the Table 2. The authentication of the samples was identified by Deputy director ZengXi Guo according to the morphological features, and the voucher specimens were deposited in the Zhejiang Institute for Food and Drug Control.

### 3.2. Screen of the Chromatographic Elution Program

The HPLC analysis was performed on a Dionex UltiMate 3000 LC series–diode array detector (DAD) system with a quaternary pump and an autosampler that could thermostat samples. Separation was achieved on Kromasil 100-5C18 (4.6 mm × 250 mm, 5 µm, Agilent Technologies, Wilmington, DE, USA). The detection wavelength was set at 235 nm for HPLC fingerprinting, and 220 nm and 280 nm for HPLC quantification. The injection volume was 20 µL.

The chromatographic separation was performed using acetonitrile (solvent A) and water (solvent B) as mobile phase at a flow rate of 1 mL·min^−1^. The gradient program was set as follows: 0–35 min: 5% A–65% A; 35–40 min: isocratic 65% A; 40–45 min: 65% A–5% A; 45–50 min: 5% A–80% A.

### 3.3. System Suitability Test of HPLC Fingerprint

Under the above conditions, resolution, and theoretical plates, the peak was calculated. Theoretical plates according to atractylone peak should not be less than 20,000, and the resolution between peak 16 and peak 17 should not be less than 1.0.

### 3.4. System Suitability Test of HPLC Quantification

Under the above conditions and theoretical plates, the peak of atractylone was calculated. It should not be less than 20,000.

### 3.5. Establishment of the HPLC Chromatogram of AMR

To establish the representative HPLC chromatogram, 16 batches of AMR as well as 11 batches of AMR stir-fried with bran were analyzed under the optimized HPLC conditions. Then, 17 common peaks were symbolized, among which four peaks were identified, namely atractylenolide III (P2), atractylenolide II (P6), atractylenolide I (P18), and atractylone (P14), as shown in Figure 1.

### 3.6. Validation of the HPLC Fingerprint Method

According to the guidelines for analytical method validation in Pharmacopeia of People’s Republic of China (volume IV) (version 2020), the precision of the HPLC method was evaluated by sampling the replicated sample solution (S9) six times with successive injections. The stability test was determined by analyzing the same standard solution at 0, 2, 4, 8, 10, 16, and 24 h. The repeatability was determined by preparing six sample solutions (S9) independently and calculating the RSDs of relative peak area and relative peak retention time.

### 3.7. Validation of the HPLC Quantification Method

According to the guidelines for analytical method validation in Pharmacopeia of People’s Republic of China (volume IV) (version 2020), the linearity regression curves for each component were obtained by plotting the peak areas (y) against the concentrations of each component standard solution. The precision of the HPLC method was evaluated by sampling each replicated component standard solution of the same sample six times with successive injections. The stability test was performed by analyzing the same standard solution at 0, 2, 5, 8, 15, 20 and 24 h. The repeatability was determined by preparing six sample solutions (S9) independently and calculating the RSDs of the contents. The recovery test was conducted to evaluate the accuracy of this method. The recovery test of a solution was performed by adding a known amount of atractylenolide III, atractylenolide II, atractylenolide I, and atractylone standard solution, respectively, to the six AMR solutions (S9) and calculating the recovery rate and the RSDs.

### 3.8. Statistical Analysis

The HPLC chromatograms of 27 batches of AMR samples were analyzed by “the similarity evaluation system for chromatographic fingerprint of TCM” software (version 2012). Chemical pattern recognition analysis was performed with SIMCA-P software (version 14.0).

## 4. Conclusions

The proposed HPLC fingerprinting and multi-component content determination method for AMR is easy to carry out with a sound level of repeatability and a high level of reliability. It provides a scientific basis for the comprehensive quality evaluation of AMR. As shown in the experimental results, the overall chemical compositions of the 27 batches of AMR processed by the two different processing methods were not identical. The similarity analysis was performed using the established HPLC fingerprint. The results showed that the 27 batches of sample had a high level of similarity. The similarity of the 17 selected common peaks was above 0.90. This study combined HPLC fingerprinting with quantitative analysis of multi-components, and PCA revealed that there were significant differences in AMR quality from different processing methods. Peaks 16, 14 (atractylone), 15, 13, 17, and 6 (atractylenolide I) were screened out by combining the CA, PCA, and LS-DA methods, which contributed to the differentiation between the raw and stir-baked AMR. In addition, the content levels of four components in AMR (atractylenolide I, II, III, and atractylone) were identified and compared with the spectra and retention times of the standards. The aim was to explore further the medicinal value of AMR and expand its clinical application. In conclusion, a quality evaluation method for AMR was established in this study to provide a reference for the quality control and clinical applications of AMR.

## Figures and Tables

**Figure 1 molecules-26-07124-f001:**
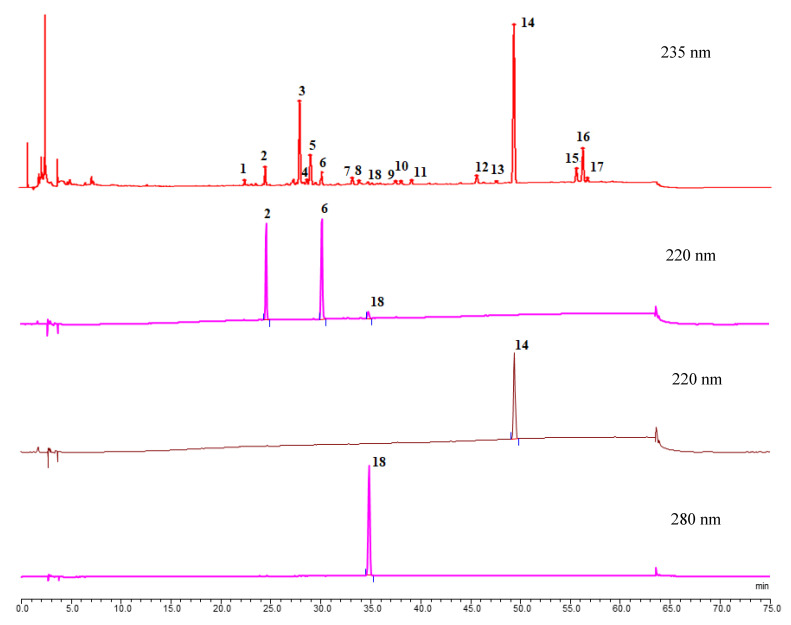
HPLC chromatogram reference fingerprint of AMR (235 nm). The identified peaks of fingerprints were in the order of atractylenolide III (2) (220 nm), atractylenolide II (6) (220 nm), atractylenolide I (18) (280 nm), atractylone (14) (220 nm).

**Figure 2 molecules-26-07124-f002:**
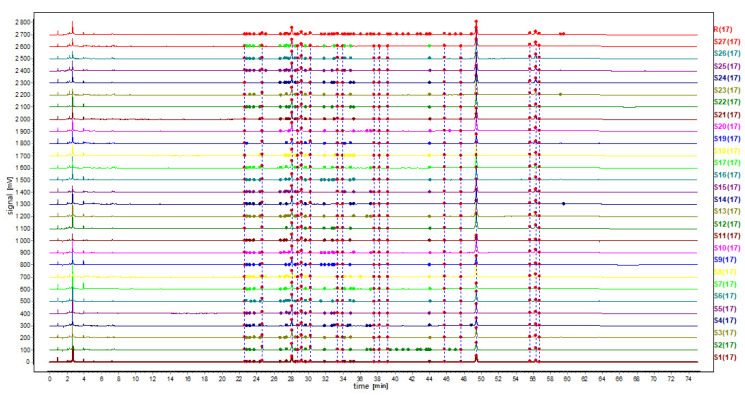
HPLC fingerprint of samples.

**Figure 3 molecules-26-07124-f003:**
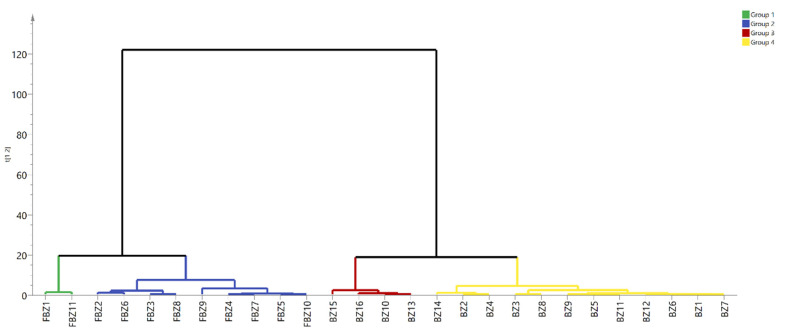
PCA dendrogram of 27 batches of samples.

**Figure 4 molecules-26-07124-f004:**
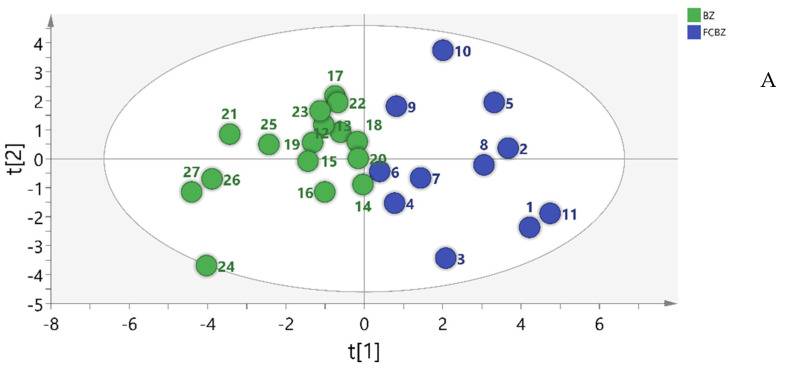
Principal component load diagram. (**A**) PCA score chart; (**B**) PCA-3D score chart.

**Figure 5 molecules-26-07124-f005:**
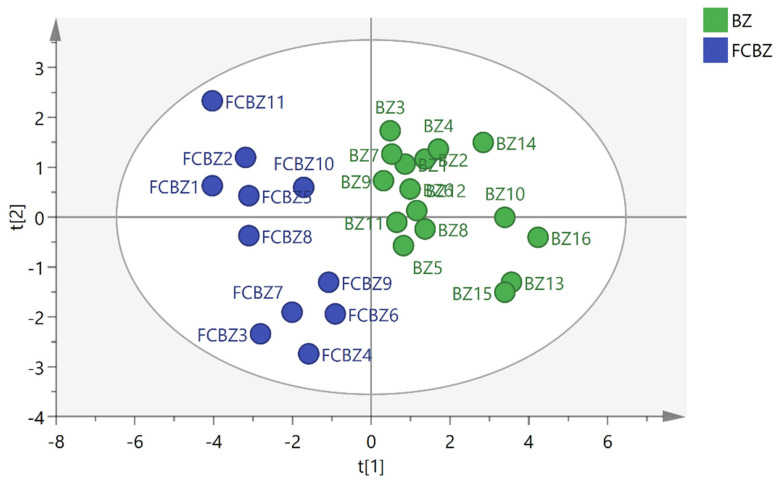
PLS-DA diagram of 27 batches of AMR sample.

**Figure 6 molecules-26-07124-f006:**
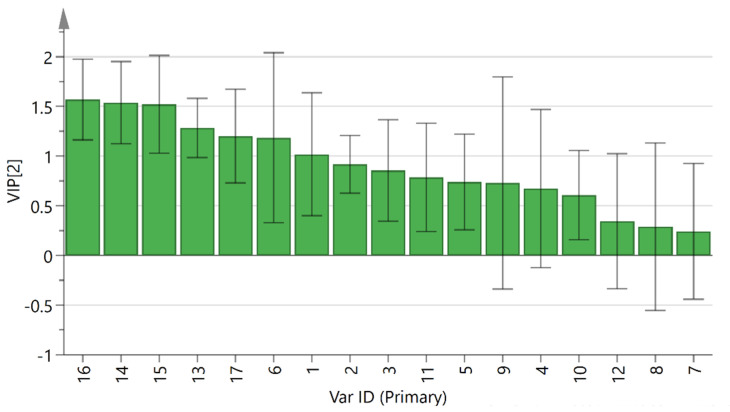
VIP diagram of 27 batches of AMR sample.

**Figure 7 molecules-26-07124-f007:**
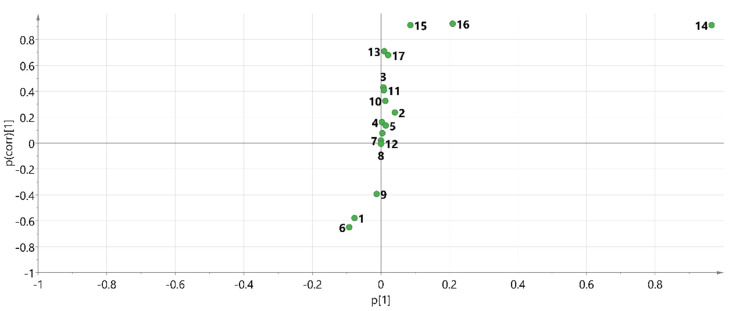
S-plot diagram of 27 batches of AMR sample.

**Table 1 molecules-26-07124-t001:** The precision, repeatability, and stability of the common peaks in AMR.

Peak No.	Precision				Repeatability				Stability			
	RRT	RSD (%)	RPA	RSD (%)	RRT	RSD (%)	RPA	RSD (%)	RRT	RSD (%)	RPA	RSD (%)
1	0.455	0.41%	0.020	0.89%	0.456	0.52%	0.020	1.93%	0.457	1.32%	0.020	2.40%
2	0.498	0.26%	0.198	0.92%	0.498	0.43%	0.198	1.98%	0.498	1.12%	0.198	1.73%
3	0.568	0.31%	0.428	1.09%	0.567	0.44%	0.428	2.04%	0.568	1.43%	0.424	1.35%
4	0.589	0.40%	0.171	1.04%	0.582	0.19%	0.171	2.15%	0.590	1.34%	0.171	2.30%
5	0.598	0.78%	0.044	1.02%	0.589	0.22%	0.044	2.15%	0.600	1.20%	0.041	1.94%
6	0.613	0.62%	0.174	0.89%	0.611	0.19%	0.174	1.88%	0.612	1.12%	0.173	1.73%
7	0.675	0.33%	0.059	1.18%	0.674	0.21%	0.059	1.92%	0.674	1.70%	0.060	1.58%
8	0.686	0.39%	0.035	1.03%	0.685	0.85%	0.035	1.95%	0.688	1.23%	0.036	2.07%
9	0.762	0.25%	0.027	1.04%	0.761	0.25%	0.027	2.07%	0.761	1.60%	0.027	1.56%
10	0.773	0.29%	0.034	1.22%	0.772	0.20%	0.034	1.88%	0.772	1.07%	0.033	0.74%
11	0.793	0.24%	0.025	1.14%	0.793	0.23%	0.025	1.96%	0.793	1.62%	0.025	1.51%
12	0.926	0.24%	0.060	0.94%	0.925	0.24%	0.060	1.95%	0.925	1.71%	0.060	1.27%
13	0.965	0.25%	0.010	1.17%	0.964	0.19%	0.010	2.06%	0.964	1.02%	0.009	1.59%
14	1.000	0.00%	1.000	0.00%	1.000	0.00%	1.000	0.00%	1.000	0.00%	1.000	0.00%
15	1.126	0.27%	0.087	1.23%	1.125	0.19%	0.087	1.94%	1.126	2.28%	0.091	1.07%
16	1.137	0.25%	0.205	0.89%	1.139	0.17%	0.205	1.96%	1.139	1.25%	0.203	1.86%
17	1.149	0.39%	0.020	1.04%	1.157	0.30%	0.020	2.03%	1.149	1.34%	0.020	0.93%

**Table 2 molecules-26-07124-t002:** The detailed information on AMR and the results of similarity.

Number	Batch Number	Origin	Chemical Pattern Recognition Analysis Number	Similarity
S1	180808	Zhejiang	FBZ1	0.991
S2	190701	Zhejiang	FBZ2	0.974
S3	19123001	Zhejiang	FBZ3	0.887
S4	200213	-	FBZ4	0.977
S5	2190921T	Zhejiang	FBZ5	0.980
S6	1912045	Zhejiang	FBZ6	0.983
S7	191114	Zhejiang	FBZ7	0.983
S8	2002013	Zhejiang	FBZ8	0.938
S9	1912176	Anhui	FBZ9	0.978
S10	2001013	Zhejiang	FBZ10	0.932
S11	20200301	Zhejiang	FBZ11	0.981
S12	190501	Zhejiang	BZ1	0.997
S13	1912043	Zhejiang	BZ2	0.993
S14	200101	Anhui	BZ3	0.991
S15	4012-190801	-	BZ4	0.993
S16	200525	Zhejiang	BZ5	0.998
S17	200103	Anhui	BZ6	0.997
S18	200201	Zhejiang	BZ7	0.998
S19	200316	Zhejiang	BZ8	0.999
S20	191001	Anhui	BZ9	0.998
S21	CP-058-200201	-	BZ10	0.995
S22	190801	Anhui	BZ11	0.997
S23	2003011	Zhejiang	BZ12	0.998
S24	200316	Zhejiang	BZ13	0.993
S25	191102	Anhui	BZ14	0.989
S26	200228	Anhui	BZ15	0.996
S27	200429	Zhejiang	BZ16	0.991

**Table 3 molecules-26-07124-t003:** Linearity equations, correlation coefficients, and linearity ranges.

Components	Linearity Equations	Correlation Coefficients	Linearity Ranges
Atractylenolide I	*y* = 0.0611 *x* − 0.0784	1.000	17.73~177.32
Atractylenolide II	*y* = 0.0490 *x* − 0.0611	1.000	17.50~175.02
Atractylenolide III	*y* = 0.0326 *x* − 0.0358	1.000	18.57~185.71
Atractylone	*y* = 0.0134 *x* − 0.0201	1.000	52.70~527.00

**Table 4 molecules-26-07124-t004:** The recovery rate of four compounds.

Components	Original (µg)	Added (µg)	Found (µg)	Recovery Yield (%)	RSD (%)
Atractylenolide I	75.71	106.38	164.08	90.11	1.33
	75.44	106.38	165.69	91.13	
	73.47	106.38	162.55	90.38	
	78.20	106.38	171.84	93.10	
	73.39	106.38	162.75	90.53	
	74.09	106.38	166.77	92.41	
Atractylenolide II	108.85	105.00	220.96	103.32	2.00
	107.75	105.00	222.69	104.67	
	112.93	105.00	218.69	100.35	
	111.30	105.00	229.84	106.26	
	105.02	105.00	218.47	104.02	
	106.24	105.00	222.94	105.54	
Atractylenolide III	114.51	111.42	229.91	101.76	1.13
	115.05	111.42	232.98	102.87	
	110.16	111.42	229.80	103.71	
	116.67	111.42	229.03	100.41	
	109.23	111.42	226.08	102.46	
	111.24	111.42	229.44	103.05	
Atractylone	1458.00	190.90	1467.60	89.00	1.62
	1459.00	190.90	1479.89	89.70	
	1416.00	190.90	1459.33	90.82	
	1510.78	190.90	1488.96	87.50	
	1429.06	190.90	1460.46	90.15	
	1424.00	190.90	1480.53	91.68	

**Table 5 molecules-26-07124-t005:** The contents of four compounds.

Sample Number	Atractylenolide III (%)	Atractylenolide II (%)	Atractylenolide I (%)	Atractylone (%)
S1	0.042	0.023	0.036	0.475
S2	0.021	0.014	0.015	0.382
S3	0.050	0.016	0.039	0.402
S4	0.039	0.016	0.031	0.549
S5	0.018	0.011	0.016	0.433
S6	0.029	0.016	0.020	0.575
S7	0.037	0.018	0.022	0.562
S8	0.026	0.018	0.023	0.331
S9	0.017	0.010	0.011	0.492
S10	0.001	0.006	0.007	0.774
S11	0.035	0.022	0.036	0.534
S12	0.010	0.011	0.009	0.740
S13	0.001	0.008	0.006	0.833
S14	0.021	0.013	0.013	0.857
S15	0.013	0.009	0.009	0.959
S16	0.028	0.015	0.020	0.826
S17	0.007	0.007	0.004	0.745
S18	0.018	0.010	0.010	0.688
S19	0.027	0.010	0.015	0.739
S20	0.018	0.017	0.011	0.810
S21	0.004	0.008	0.003	0.981
S22	0.012	0.011	0.006	0.673
S23	0.009	0.009	0.007	0.763
S24	0.040	0.015	0.011	1.175
S25	0.006	0.008	0.004	0.991
S26	0.021	0.011	0.014	0.979
S27	0.015	0.009	0.010	1.080

## Data Availability

Data are contained within the article.

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
