# Peer review of "Quality Evaluation of Atractylodis Macrocephalae Rhizoma Based on Combinative Method of HPLC Fingerprint, Quantitative Analysis of Multi-Components and Chemical Pattern Recognition Analysis"

_molecules, 2021, doi:10.3390/molecules26237124_

Round 1

Reviewer 1 Report

Dear Authors

In my opinion, the topic is interesting to certain readers in Molecules. There is substantial work done on the topic. However, the following are a few comments which should be answered before a final recommendation.

  1. Please give more detailed information about your research novelty
  2. Did you perform system suitability on your HPLC quantification method? if yes, please write in your paper, if not I suggest you do that.
  3. Did you use a QC sample in your HPLC fingerprint analysis? Please write if you used it. If not, why did not use the QC sample?
  4. Fig 1, HPLC chromatogram of Dalbergiae odoriferae? I think odoriferae is not your sample right?
  5. In Fig 2, the peak is not clear in the chromatogram, could you make it bigger so we can see the peaks
  6. Besides VIP, you should report the S-plot to give a more comprehensive prediction on the variable that affected the groping samples

Reviewer 2 Report

There is no discussion in the article, no comparison with other publications where the compounds were marked. Sloppy description of the methodology.

  1. In Fig. 1 authors signed chromatogram A as Dalbergia odorifera lignum whether it was a mistake in the caption or a wrong chromatogram was added?
  2. In material and method section:

3.2 full names of equipment and manufacturers should be added. What does it mean elution A, do authors mean eluent?

3.3. how were the 4 basic compounds identified, just by retention times and spectra? How authors are sure that they are these compounds,  not the derivatives of the compounds tested? In my opinion, the authors should show the spectra of these 4 components and compare them with the spectra of the standards.

There are too many pharmacopeia in “pharmacopeia”

There are also a lot of errors in some words even in abstract (l.16 contends)

Round 2

Reviewer 2 Report

The article has been improved, the authors added the spectrum of compounds in response to the previous review. My suggestion would be to add them to the supplementary materials. 

But there are still some things which should be improved:

Please sign on the first chromatogram (A) which peak corresponds to which standard.  If I can guess it in the case of peaks 1 and 4, I have doubts about the other two peaks. Please number all 17 peaks in question.

In: 3. Materials and methods; 3.2. 

please finish this sentence “Acetonitrile (component A) and water (component B) in a gradient elution” 

In some cases the name of the compound is atractylone and in others it is atractylon, which the authors believe is correct?

There are still some language errors like:

Some sholars have also conducted exploratory work on the fingerprinting of AMR of different origins 

Manwhile, the corresponding limits have not been established so far

Therefore, the present study aims to investigate HPLC fingerprin of AMR and study on qantification of atractylenolide I, II, III, and atractylone on the bisis of chemometric analysis
